# Implementing a universal gate set on a logical qubit encoded in an oscillator

Reinier W. Heeres [1], Philip Reinhold [1,2], Nissim Ofek[1], Luigi Frunzio [1], Liang Jiang [1], Michel H. Devoret[1] & Robert J. Schoelkopf[1]

A logical qubit is a two-dimensional subspace of a higher dimensional system, chosen such that it is possible to detect and correct the occurrence of certain errors. Manipulation of the encoded information generally requires arbitrary and precise control over the entire system. Whether based on multiple physical qubits or larger dimensional modes such as oscillators, the individual elements in realistic devices will always have residual interactions, which must be accounted for when designing logical operations. Here we demonstrate a holistic control strategy which exploits accurate knowledge of the Hamiltonian to manipulate a coupled oscillator-transmon system. We use this approach to realize high-fidelity (98.5%, inferred), decoherence-limited operations on a logical qubit encoded in a superconducting cavity resonator using four-component cat states. Our results show the power of applying numerical techniques to control linear oscillators and pave the way for utilizing their large Hilbert space as a resource in quantum information processing.

[1] Departments of Physics and Applied Physics, Yale University, New Haven, Connecticut 06520, USA. [2] Present address: Quantronics group, SPEC, CEA, Gif-sur-Yvette 91191, France. Reinier W. Heeres and Philip Reinhold contributed equally to this work. Correspondence and requests for materials should be addressed to R.W.H. (email: reinier@heeres.eu)

Quantum error correction aims at the creation of logical qubits[1, 2] whose information storage and processing capabilities exceed those of its constituent parts. Significant progress has been made toward quantum state preservation by repeated error detection using stabilizer measurements in trapped ions[3, 4], nitrogen vacancy centers[5], and superconducting circuits[6–8]. In order to go beyond storage and to manipulate the encoded information, one must perform operations on the whole system in such a way that it results in the desired transformation within the two-dimensional subspace defining the logical qubit. Any encoding scheme will consist of multiple interacting components where the system dynamics are not naturally confined within the logical subspace. Therefore, implementing operations requires carefully tailored controls which address each component of the system and manage their mutual interactions. Recent efforts have achieved this level of control and have demonstrated operations on a five qubit code in nuclear spin ensembles[9] and a seven qubit code in trapped ions[4].

An alternative to logical qubit implementations based on multiple two level systems is to encode quantum information in continuous variable systems or oscillators, for which there are several schemes[10, 11]. In particular so called "cat states", which are superpositions of coherent states, can be used as the logical states of an encoded qubit[12]. They are attractive because coherent states are eigenstates of the photon annihilation operator ($\hat{a}$) and therefore single-photon loss induces simple, tractable errors[13].

Replacing several two level systems by an oscillator drastically reduces the hardware cost and complexity by requiring fewer components to fabricate and manipulate. However, introducing higher dimensional modes raises the issue of how to realize complete control over the system. Driving an isolated harmonic oscillator results in a displacement operation, which can only produce coherent states from the vacuum. Any oscillator-based logical qubit scheme will require a richer class of operations, which one can access via coupling to a nonlinear system.

In the case of a frequency-tunable qubit coupled to an oscillator with the Jaynes-Cummings (JC) interaction ($H_{JC} = \sigma_+\hat{a} + \sigma_-\hat{a}^\dagger$), it has been demonstrated that it is possible to prepare arbitrary states in the oscillator[14, 15].

In the far off-resonant case, where the JC interaction reduces to the dispersive Hamiltonian ($H_d/\hbar = \chi a^\dagger a |e\rangle\langle e|$), a small set of operations acting on a timescale of $2\pi/\chi$ is in principle sufficient for universal control[16, 17] and has been used for non-trivial operations[18, 19]. Generally, however, any approach decomposing an arbitrary operation into a sequence of elementary gates generates only a small subset of physically allowed control fields. It, therefore, suffers from two issues limiting the achievable fidelity. First, the constructed sequences may require an unacceptably large number of gates, limiting operations which are feasible in the presence of decoherence. Second, the idealized model used by a constructive approach typically fails to account for the existence of higher order Hamiltonian terms such as the Kerr non-linearity $H_{Kerr}/\hbar = \frac{K}{2}(\hat{a}^\dagger)^2\hat{a}^2$ and spurious residual couplings in multi-qubit systems.

In this work, we address these problems by considering a full model of the time dependent Hamiltonian in the presence of arbitrary control fields. Nuclear magnetic resonance experiments have shown that, if the available controls are universal, numerical optimization procedures can reliably solve the inversion problem of finding control fields to implement an intended operation. These optimal control algorithms, in particular the Gradient Ascent Pulse Engineering (GRAPE) method[20, 21], have been successfully employed in a variety of other fields[22, 23]. Since GRAPE crucially depends on the model of the system, its successful application is powerful evidence that the Hamiltonian used accurately captures the system dynamics over a broad range of driving conditions.

## Results

**System and control protocol demonstration.** The physical system used in our experiments is schematically depicted in Fig. 1a. The seamless aluminum $\lambda/4$ coax-stub cavity resonator[24] with a fundamental frequency 4452.6 MHz has an energy relaxation time of 2.7 ms. A single-junction transmon with transition frequency 5664.0 MHz and an harmonicity of 236 MHz is dispersively coupled to the oscillator, resulting in an interaction term $\chi\hat{a}^\dagger\hat{a}|e\rangle\langle e|$, with $\chi/2\pi = -2.2$ MHz. Crucially, additional higher order terms are determined accurately through a separate set of calibration experiments (Supplementary Note 1, Supplementary Fig. 1, Supplementary Table 1). Control of the system is implemented through full in-phase/quadrature modulated microwave fields centered on the transmon (oscillator) frequency and sent to the pin coupling to the transmon (oscillator) mode (setup schematic in Supplementary Fig. 2). In the rotating wave approximation, this results in the

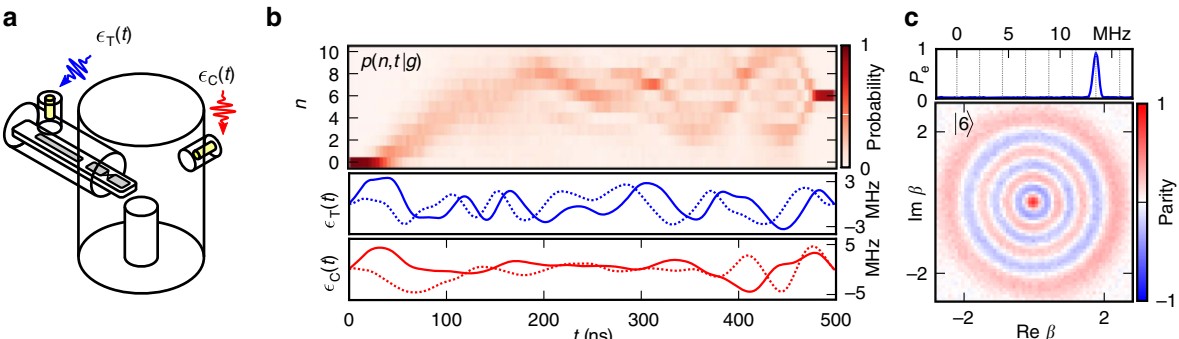

**Fig. 1** Experimental system and demonstration of control strategy. **a** Schematic drawing of the experimental system. A $\lambda/4$ coax-stub cavity resonator is coupled to a transmon and readout resonator on a sapphire substrate. Input couplers close to the transmon and cavity deliver the respective time-dependent microwave control fields $\epsilon_T(t)$ and $\epsilon_C(t)$. **b** Lower panel: optimized transmon and oscillator control waveforms of length approximately $2\pi/\chi$ to take the oscillator from vacuum to the 6-photon Fock state. Solid (dotted) lines represent the in-phase (quadrature) field component. Upper panel: oscillator photon-number population trajectory vs. time conditioned on transmon in $|g\rangle$. A complex trajectory occupying a wide range of photon numbers is taken to perform the intended operation. **c** Characterization of the oscillator state using Wigner tomography (bottom) and transmon spectroscopy (top), where grey dashed lines indicate the transition frequency associated with the first seven Fock states. The single peak in the spectroscopy data directly reveals the oscillator's population due to the dispersive interaction giving a frequency shift of $6\chi/2\pi \approx 13$ MHz

drive Hamiltonian $H_c/\hbar = \epsilon_C a + \epsilon_T \sigma_- + \text{h.c.}$ Modulation using an arbitrary waveform generator allows the coefficients $\epsilon_C$ and $\epsilon_T$ to be arbitrary complex-valued functions of time.

As an example application of GRAPE to our system (Supplementary Note 2), we find $\epsilon_C(t)$ and $\epsilon_T(t)$ such that, starting from the vacuum (Supplementary Fig. 3, Supplementary Note 3), after 500 ns of driven evolution the system ends up in the state $|g, 6\rangle$, as shown in Fig. 1b, c. This highly nontrivial operation, effectively realizing a $|6\rangle\langle 0|$ coupling term on the oscillator, is enabled by the dispersive Hamiltonian using only linear drives on the transmon and the oscillator.

**Encoding a logical cat-qubit**. Using our control strategy, we can target the creation and manipulation of a logical qubit encoded in an even-parity four-component cat subspace. Omitting normalization, the code states in this subspace can be written as

$$| \pm Z_L \rangle = |\alpha\rangle + |-\alpha\rangle \pm |i\alpha\rangle \pm |-i\alpha\rangle \tag{1}$$

where we use $\alpha = \sqrt{3}$. These code words are both of even photon number parity, and are distinguished by their photon number

modulo 4:

$$|+Z_L\rangle = \sum_n \frac{\alpha^{4n}}{\sqrt{(4n)!}} |4n\rangle \tag{2}$$

$$|-Z_L\rangle = \sum_n \frac{\alpha^{4n+2}}{\sqrt{(4n+2)!}} |4n+2\rangle \tag{3}$$

Single photon loss, the dominant error channel for the system, transforms both code words to states of odd photon number parity. The encoded information, however, is preserved by this process as long as one can keep track of the number of photons that have been lost. Since parity measurements can be performed efficiently and non-destructively[25], single photon loss is a correctable error[13].

Using GRAPE, we create a universal set of gates on our logical qubit, which includes X and Y rotations by $\pi$ and $\pi/2$, as well as Hadamard and T gates. These pulses are each 1100 ns $\approx 2.4 \times 2\pi/\chi$ in length with a 2 ns time resolution, although 99% of the spectral content lies within a bandwidth of 33 MHz (27 MHz) for the transmon (oscillator) drive (Supplementary Fig. 4). Each operation is designed to begin and end with the transmon in the ground state. Additionally, we create encode ($U_{enc}$) and decode

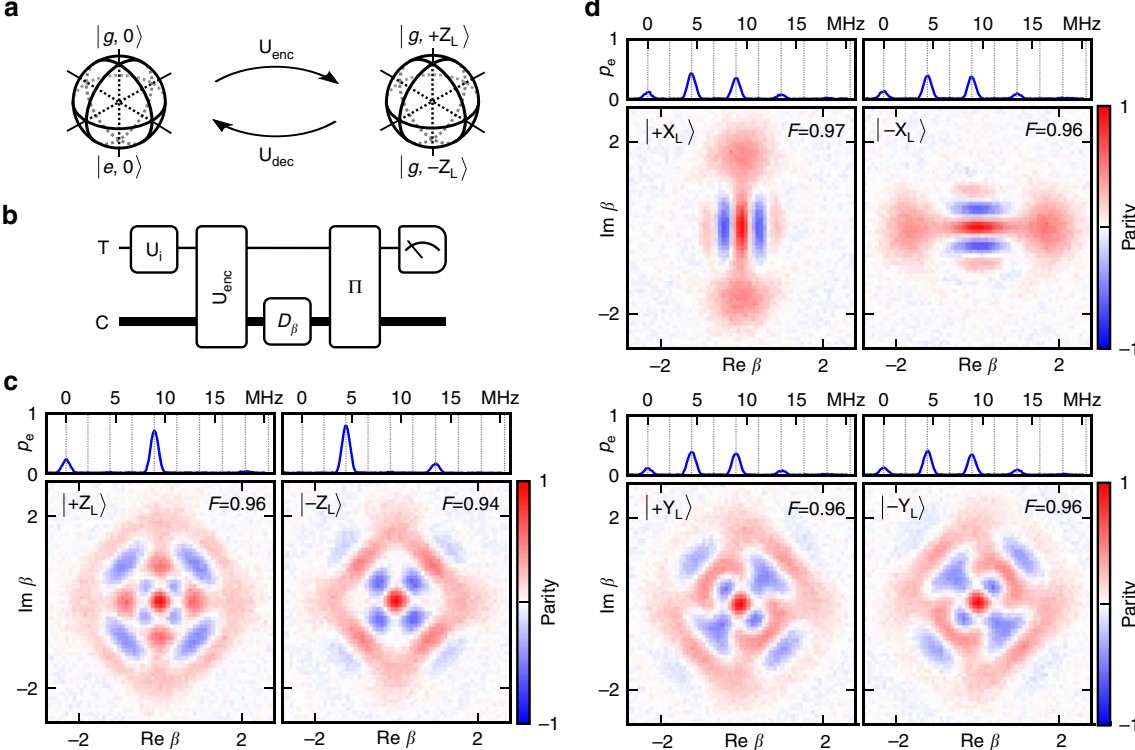

**Fig. 2** Characterization of encoded states. **a** $U_{enc}$ and $U_{dec}$ are operations that coherently map between two distinct two-dimensional subspaces, represented by Bloch spheres. The first subspace consists of the transmon $|g\rangle$ and $|e\rangle$ levels, with the oscillator in the vacuum. The second is given by the oscillator-encoded states $|+Z_L\rangle$ and $|-Z_L\rangle$ (Eq. (1)), with the transmon in the ground state. **b** Wigner tomography sequence that characterizes the encoded states. A transmon state is prepared by applying an initial rotation $U_i$ and is mapped to the oscillator using $U_{enc}$. An oscillator displacement $D_\beta$ followed by a parity mapping operation $\Pi$ (implemented using an optimal control pulse) allows one to measure the oscillator Wigner function $W(\beta)$. The transmon can be re-used to measure the oscillator's parity because the encoding pulse leaves the transmon in the ground state with high probability ($p > 98\%$). **c** Applying $U_{enc}$ to the transmon states $|g\rangle$ and $|e\rangle$ produces states whose Wigner functions are consistent with the intended encoded basis states (Eq. (1)). A transmon spectroscopy experiment (*top panel*) illustrates that only photon number states with $n = 0 \mod 4$ and $n = 2 \mod 4$ are present for logical state $|+Z_L\rangle$ and $|-Z_L\rangle$, respectively. **d** Applying $U_{enc}$ to superpositions of the transmon basis states demonstrates that the relative phase is preserved and that $U_{enc}$ is a faithful map between the transmon and logical qubit Bloch spheres. These states, on the equator of the Bloch sphere, are equally weighted superpositions of $|+Z_L\rangle$ and $|-Z_L\rangle$ and, therefore, contain all even photon numbers present in the basis states

($U_{dec}$) pulses to transfer a bit of quantum information between our transmon $\{|g, 0\rangle, |e, 0\rangle\}$ subspace, which we can easily prepare and measure, and our encoded subspace $\{|g, +Z_L\rangle, |g, -Z_L\rangle\}$ (Fig. 2a).

We characterize the encode operation by preparing all six cardinal points on the transmon Bloch sphere, applying the encode pulse and performing Wigner tomography on the oscillator (Fig. 2b–d). Maximum likelihood reconstruction of the density matrix associated with the measured Wigner functions indicates an average state fidelity of 0.96. This metric underestimates the fidelity of $U_{enc}$ because it is affected by several sources of error not intrinsic to the encoding operation itself, including error in the parity mapping and measurement infidelity.

**Gate characterization.** Process tomography provides a full characterization of a quantum operation, but depends on pre-existing trusted operations and measurements which are not available for our encoded subspace. However, an indirect characterization of a gate $U_X$ on our logical qubit can be performed using the operation $U_{dec}U_XU_{enc}$, which maps the transmon subspace onto itself. This allows one to use the trusted state preparations and measurements on the transmon to perform tomography on the composite process (Fig. 3a). The

reconstructed process matrices in Fig. 3b show qualitative agreement with the intended encoded qubit gates. The process fidelities we report are average gate fidelities $\mathcal{F}(\mathcal{E}_1, \mathcal{E}_2) \equiv \int d\psi F(\mathcal{E}_1(\psi), \mathcal{E}_2(\psi))$, where $F$ is the usual quantum state fidelity $F(\rho_1, \rho_2) = \mathrm{Tr}(\rho_1\rho_2)$. We can break the measured infidelity down into three parts: transmon preparation and measurement error, encode-decode error and gate error. Using the experimentally determined process fidelities both without any operation $\mathcal{F}_{PT}(\mathrm{No\,Op.}) = 0.982$, as well as with the encode and decode pulses $\mathcal{F}_{PT}(U_{dec}\,U_{enc}) = 0.964$, we estimate an infidelity contribution of approximately 1.8% for each of the first two components. To account for these factors to first order, the infidelity of operations on the encoded qubit are reported relative to $\mathcal{F}_{PT}(U_{dec}U_{enc})$. We find an average infidelity of 0.75% over our set of nine gates (Table 1).

In order to establish the fidelity of this set of operations more accurately, we perform randomized benchmarking[26] (RB) on our encoded qubit (Fig. 4a). Careful analysis is required to infer the actual gate fidelity, as leakage out of the logical space in the oscillator does not present itself directly in the binary measurement of the state of the transmon qubit. Simulations show that such an RB experiment on a logical qubit with a larger associated Hilbert space typically overestimates the fidelity by a factor $1.7 \pm 0.1$ (Supplementary Fig. 6, Supplementary Note 4). From the resulting data (Fig. 4c) we infer an average gate fidelity

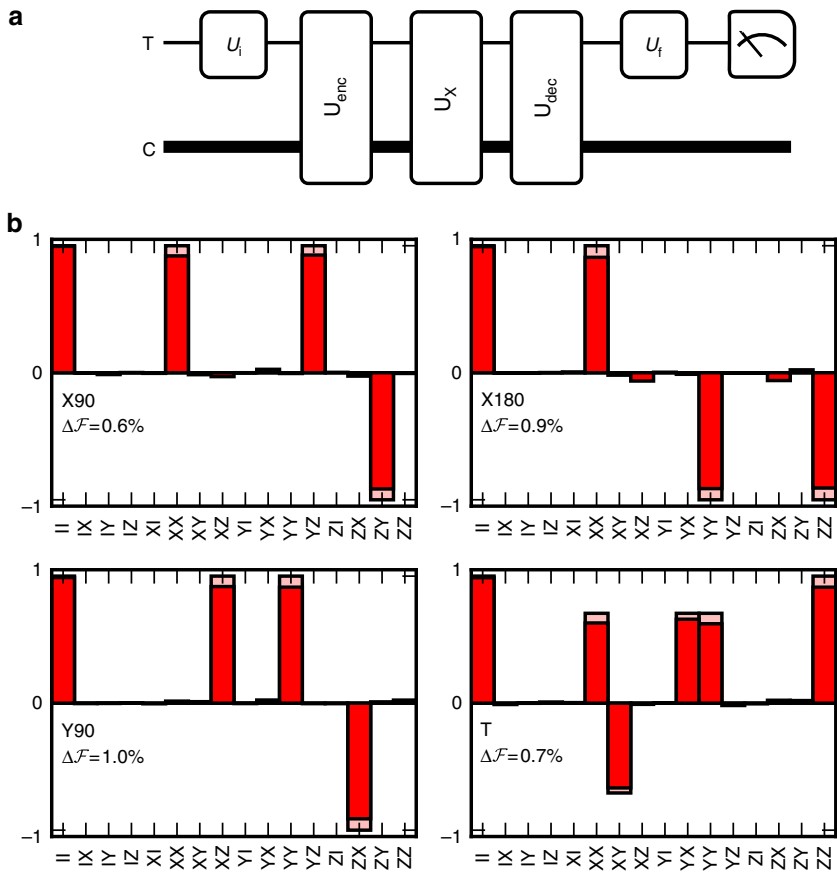

**Fig. 3** Process tomography of operations on encoded qubit. **a** In order to characterize a gate $U_X$ on the encoded qubit, transmon process tomography is performed on the operation $U_{dec}U_XU_{enc}$. Process tomography is implemented by performing an initial transmon rotation $U_i$ right after state preparation (Supplementary Note 3), as well as a final transmon rotation $U_f$, right before measurement of the transmon. **b** Process tomography results for selected operations (for additional operations, see Supplementary Fig. 5). The process tomography yields an estimated quantum channel G. We represent this channel in the Pauli transfer representation. The bar labeled with operators $AB$ ($A$, $B \in \{I,X,Y,Z\}$, the Pauli basis matrices) represents $\mathrm{Tr}(AG(B))/2$. Red and pink bars indicate the experimental and ideal values, respectively. The infidelity $\Delta\mathcal{F}_{PT}$ of operation $U_X$ is estimated as the difference between $\mathcal{F}_{PT}(U_{dec}U_XU_{enc})$ and $\mathcal{F}_{PT}(U_{dec}U_{enc}) = 0.964$. The selected set of operations, {X180,X90,Y90,T}, allows universal control of the logical qubit

of 0.985. This measurement is so sensitive to the quality of the applied gates that it is the ideal metric to use to fine-tune several experimental parameters (Supplementary Fig. 7, Supplementary Fig. 8, Supplementary Note 5). The infidelity of each of the individual gates is isolated using interleaved randomized benchmarking[27] (iRB), which alternates between a single fixed and a random gate (Fig. 4b). Comparing the fitted decay constants of the RB and iRB results allows us to extract the fidelity of the fixed gate. The results are summarized in Table 1, together with the gate fidelities based on process tomography (Fig. 3) and Lindblad master equation simulations accounting for finite $T_1$ and $T_2$ of the transmon and oscillator. We note that all gates are implemented with an approximately equal infidelity of 1.5%. The results obtained using process tomography and iRB are not consistent, leading us to conclude that the approximation of the infidelity as the difference between $\mathcal{F}_{PT}(U_{dec}U_X U_{enc})$ and $\mathcal{F}_{PT}(U_{dec}U_{enc})$ leads to an underestimation of the underlying infidelity of $U_X$. While several sources of decoherence are accounted for in the master equation simulations, the dominant source of infidelity in the model is transmon dephasing ($T_2 \approx 43\,\mu s$). The good agreement between simulations and experiment indicates that the infidelity is primarily caused by decoherence and that additional contributions associated with imperfections in the model Hamiltonian and the applied pulses are a significantly smaller effect.

## Discussion

A logical quantum bit consists of a quantum system with multiple degrees of freedom that can be used to correct for a finite set of errors, at the expense of being more complicated to control. Before one can realize the ultimate goal of robust and high fidelity operations that surpass the performance of the physical qubits, there are several challenging steps which must be demonstrated. First one must devise a code that can detect and correct for the dominant errors, second is to implement the code by demonstrating encoding and decoding operations, third is to show the ability to detect and correct the errors, and fourth is to manipulate the information in the encoded system by performing logical operations. Finally, one must eventually combine *all* of these components to improve the systems lifetime and the fidelity of operations.

In this work, we show an important step along this path, namely the first manipulations of a logical qubit encoded in cat-states. At this level one should generally expect the overall error rate to actually increase relative to the physical components. This is due to the additional overhead of implementing a logical qubit, which originates from the redundant encoding required to detect and correct errors. For instance, the seven qubits in the Steane code[28] result in an effective error rate, which is larger by a factor 7. Our scheme, the cat-code using several levels in an oscillator, increases the error rate by a factor $\bar{n} \approx 3$ in the best case scenario.

The increased system complexity of a logical qubit poses additional challenges and sources of errors when performing logical operations. First of all, implementing any of the operations is non-trivial as the logical basis states are typically not easily constructed using the available controls. In fact, this is a key property of the encoding as it prevents the information to decohere due to interaction with the environment. When starting from the system's ground state, for example, a sequence of several single- and two-qubit gates is required to produce a logical basis state in the Steane code, or a complex pulse in our scheme. Additionally, when integrating many physical systems it remains

**Table 1 Operation fidelities**

| Gate | $1-\mathcal{F}_{RB}$ (%) | $\Delta\mathcal{F}_{PT}$ (%) | $1-\mathcal{F}_{sim}$ (%) |
|---|---|---|---|
| I | $0.78 \pm 0.06$ | 0.51 | 0.44 |
| X90 | $1.34 \pm 0.09$ | 0.57 | 1.20 |
| -X90 | $1.54 \pm 0.10$ | 0.71 | 1.31 |
| X180 | $1.89 \pm 0.12$ | 0.88 | 1.57 |
| Y90 | $1.63 \pm 0.11$ | 0.98 | 1.23 |
| -Y90 | $1.38 \pm 0.09$ | 0.52 | 1.23 |
| Y180 | $2.18 \pm 0.14$ | 0.99 | 2.18 |
| H | $1.58 \pm 0.11$ | 0.86 | 1.52 |
| Average | $1.54 \pm 0.10$ | 0.75 | 1.34 |
| $U_{enc}U_{dec}$ | $2.89 \pm 0.18$ | 1.39 | 2.71 |
| T | – | 0.71 | 0.66 |

Measured and simulated gate infidelities. $\mathcal{F}_{RB}$, $\Delta\mathcal{F}_{PT}$ and $\mathcal{F}_{sim}$ are the values extracted from interleaved randomized benchmarking, process tomography (see Fig. 3) and simulations using the Lindblad master equation, respectively. The row labeled "average" gives the fidelities averaged over the first eight gates, which is the set used in the standard randomized benchmarking experiment

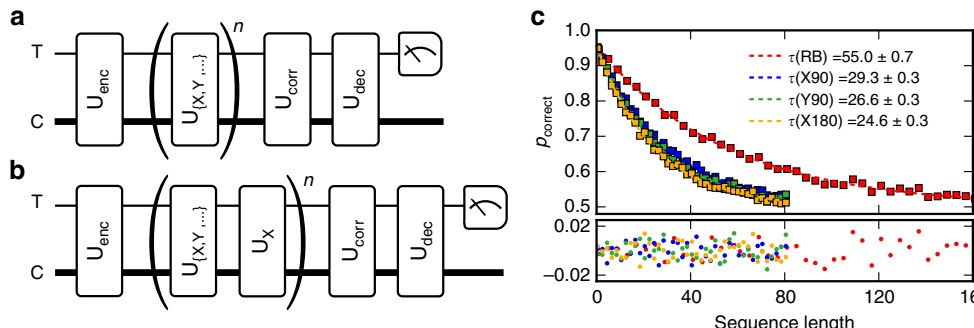

**Fig. 4** Randomized benchmarking of operations on encoded qubit. **a** Randomized benchmarking (RB) sequence. In RB a sequence of Clifford operations of length $n$ is chosen at random ($U_{\{X,Y,...\}}$), followed by the operation which inverts the effect of the sequence ($U_{corr}$). In order to apply this technique to the operations on the encoded qubit, we begin the experiment by encoding, and decode before measurement. Our implementation of RB creates a new random gate sequence for every measurement, and is thus not biased by the distribution of sequences which are measured. **b** Interleaved randomized benchmarking (iRB) sequence: In order to establish the fidelity of a single operation (here, $U_X$), the operation is interleaved with random operations, and the benchmarking result is compared with the non-interleaved case. **c** The probability of measuring the correct result vs. sequence length $n$ is fit to a two parameter model $p_{correct} = 0.5 + Ae^{-n/\tau}$. The *lower panel* shows the fit residuals. Each data point is the result of 2000 averages, with a new sequence realization every shot. The error averaged over all gates is computed as $r = c(1-e^{-1/\tau(RB)})/2$[26]. The average error for a single gate $X$ is computed as $r(X) = c(1-e^{1/\tau(X)-1/\tau(RB)})/2$[27]. The factor $c = 1.7 \pm 0.1$ is a correction factor to compensate for the underestimation of the error rate in the presence of leakage to a larger Hilbert space (Supplementary Note 4)

an open experimental question whether control-field cross-talk and inter-qubit interactions, which introduce coherent and correlated errors, can be engineered sufficiently small. Therefore, demonstrating accurate manipulation of a logical qubit is an important and necessary first step toward error-corrected quantum computation. Such experiments lead to a better understanding of experimental nonidealities and other sources of errors.

By mapping a bit of quantum information from the transmon onto cat-states in the oscillator, we have transferred the information onto a system with a coherence time which is more than an order of magnitude larger (Supplementary Fig. 9). The overhead of encoding/decoding is approximately 3%, and, therefore, it is beneficial when storing a state for more than around 3% of the transmon coherence time, or 1.3 µs. The dispersive coupling to the transmon still allows the information encoded in the logical qubit to be manipulated directly, albeit with a fidelity smaller than that of transmon operations. There are a number of protocols where this trade-off between operation fidelity and lifetime is desirable, such as entanglement distillation[29] and quantum repeaters[30]. Photon-loss error correction can enhance the lifetime of the logical cat-code qubit even further[13], but would not enhance the operation fidelity. The reason is twofold: first of all, the system is not in a cat-state during the gate operation (Supplementary Fig. 10), and, second, photon-loss is a much less significant source of errors than transmon dephasing. Although improving the transmon coherence time would directly result in higher fidelity gates, it is likely that this will remain the dominant source of errors.

However, simulations show that a large fraction of the errors that occur during an operation are detectable (Supplementary Table 2), and could, therefore, be mitigated using erasure correcting codes[31]. This originates from the fact that the oscillator-transmon system takes a complicated trajectory through its Hilbert space during an operation. An error drastically alters this trajectory and, therefore, the final state at the end of the operation. For example, in approximately 50% of the cases an error occurs, the transmon will be left in the excited state. Additionally, it is unlikely that the oscillator will remain in the logical subspace after an error, and if this could be measured efficiently it would imply that, for the designed pulses, approximately 95% of the errors are detectable.

It might be possible to increase the fraction of detectable errors by optimizing control pulses in the presence of decoherence, an established technique[32], combined with an appropriate modification the cost function; we have not yet thoroughly explored this approach. A more fundamental open question is whether the GRAPE algorithm can be used to design pulses which implement operations fault-tolerantly (e.g., with respect to transmon dephasing).

In conclusion, we have demonstrated a high-fidelity implementation of a universal set of gates on a qubit encoded into an oscillator using the cat-code. The low error rates for these operations are verified using both process tomography and randomized benchmarking, and the results are consistent with simulations which account for decoherence. We obtained these operations by numerically optimizing time-dependent drives which make use of the well-characterized dispersive interaction between the far detuned oscillator and transmon modes. While in this Article we have focused on realizing and characterizing single-qubit operations on cat-encoded qubits, this control technique is not restricted to these goals, and is in principle capable of crafting arbitrary unitary operations on the transmon-oscillator system. The high quality of these operations depends critically on an accurate characterization of the system

Hamiltonian, and demonstrates the utility of numerical optimal control for realizing quantum information processing.

**Data availability**. Relevant data is available from R.W.H. upon request.

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

## Acknowledgements
We would like to thank Katrina Sliwa and Michael Hatridge for providing the parametric amplifier, Chris Axline, Jacob Blumoff, Kevin Chou, and Chen Wang for discussions regarding sample design, Stefan Krastanov, Chao Shen and Victor Albert for discussions on universal control and Steve Flammia and Robin Blume-Kohout for advice about tomography. This research was supported by the U.S. Army Research Office (W911NF-14-1-011). P.R. was supported by the U.S. Air Force Office of Scientific Research (FA9550-15-1-0015), L.J. by the Alfred P. Sloan Foundation and the Packard Foundation. Facilities use was supported by the Yale Institute for Nanoscience and Quantum Engineering (YINQE), the Yale SEAS cleanroom, and the National Science Foundation (MRSECDMR-1119826).

## Author contributions
R.W.H. and P.R. performed the experiment and data analysis under the supervision of R.J.S. P.R. developed the GRAPE implementation, which created the optimal control pulses. N.O. developed the Field Programmable Gate Array hardware, which controls the experiment. M.H.D. and L.J. provided theoretical support. R.W.H. and L.F. fabricated the transmon qubit. R.W.H., P.R., and R.J.S. wrote the manuscript with contributions from all authors.

## Additional information

**Competing interests:** R.J.S., M.H.D., and L.F. are equity holders and consultants at Quantum Circuits, Inc. R.W.H., P.R., L.J., L.F., and R.J.S. are co-inventors on a patent submission by Yale University related to this work. N.O. declares no competing financial interest.

