## [Peer Review file · Nature Communications]

Reviewer #1 (Remarks to the Author):

This paper describes the experimental implementation of a universal set of single qubit gate operations on a logical qubit, encoded in a superposition of even cat states of a microwave cavity. Such a logical qubit encoding has been called 'cat code' and in previous publications the authors have established that it can be used to correct for photon loss, the dominant decoherence mechanism in cavities. Compared to other encodings of logical qubits, cat codes require much less hardware resources and therefore appear very realistic.

The paper demonstrates experimentally the next step towards using logical cat-code qubits by demonstrating initialization from a transmon qubit and a universal set of single qubit gates. To achieve these operations the authors use optimal control strategies developed originally for NMR.

These are beautiful results, technically sound and well presented. Operating logical qubits is certainly a major step, which makes them, in principle, worth being published in Nature Communications.

However, while the reported gate fidelities of approximately 99% are sufficient for physical qubits as part of some error correction schemes, they are unacceptably low for a useful logical qubit. Presenting the results in the light of quantum error correction and logical qubits suggests that these gates are compatible with the cat code error correction schemes and benefit from them in terms of fidelity, but the authors do not provide any information supporting this suggestion nor do they emit any reservations. The current presentation of the paper in the light of quantum error correction is therefore misleading.

To remove this issue the authors should either clearly state that their gates cannot be protected with the cat-code error correction scheme and that their fidelity therefore has to be taken as is. Or they should give avenues how fidelities can be improved by correcting for the dominant error mechanisms. This will allow the reader to judge whether their approach is actually a huge step towards scalable quantum computation using cat codes, or just a beautiful side road.

I see at least two issues with correcting errors in these gates:

- The authors mention in their discussion of Table I that they can explain most of the infidelity by transmon dephasing. It is not obvious how this error could be corrected in the cat-code scheme, which would suggest that using a cat-code qubit does not present any benefits over directly using a transmon. Even worse, as the gates take much longer they may be affected even more by dephasing. Can the pulses be modified for the gates to be insensitive to transmon dephasing by adding appropriate cost functions to grape?

- It is not obvious that the resonator stays in a superposition of even cat states during the gate so the previously proposed cat-code correction scheme might not work during the gate and it is not obvious if single photon loss during the gate can be corrected afterwards or is transformed into uncorrectable errors by the action of the drives. The authors might want to show resonator state evolution during the gate in the supplementary material.

In addition I have a few more minor issues:

- How does the scheme scale to larger cats which are required for the cat code correction scheme to be most effective?

- Figure 3b-c: The x axis labels should be identified more clearly

- The definition of fidelity should be given in the text when it is first used instead of the caption of

table I.

Reviewer #2 (Remarks to the Author):

The authors describe experiments to demonstrate universal control of a protected logical qubit encoded in the state of a bulk microwave cavity dispersively coupled to a transmon qubit. The encoding scheme is the cat-code previously described by this group. Tools from optimal control theory are used to generate high-fidelity encoding and gate pulses; success of this approach requires accurate knowledge of higher-order terms in the qubit-cavity Hamiltonian. The experiments are beautiful and showcase an unprecedented level of control over an extremely complex multipartite quantum system. However, the current presentation raises obvious questions about the viability of the described approach as a means to realizing universal fault-tolerant quantum computation. It would be extremely useful for the community to have the authors address head-on some of the apparent limitations of the proposed approach, so that readers can judge how the cat-code approach to scalable quantum information processing with superconductors compares to more "conventional" approaches based on planar integrated circuits and the surface code.

Specifically:

1. Given that the fidelity of encoding in the cat code is less than the fidelity of gate operations on the bare (unprotected) transmon qubit, it would seem that operations in the protected subspace will always lead to lower fidelity than operations on the bare transmon, provided that one takes into account the penalty associated with encoding and decoding. Is there an understanding of the sources of error that contribute to infidelity of the encoding operations and a path to reducing this infidelity?

2. The cat-code is motivated by the observation that cavity photon loss is the dominant source of decoherence in this system; therefore, a code that allows one to monitor and correct this loss provides a high degree of protection. However, in their analysis of the fidelity of operations in the encoded subspace, the authors find that dephasing of the transmon is the dominant source of fidelity loss. A transmon dephasing time of 43 microseconds is quite respectable; at what point will transmon dephasing have negligible impact on the fidelity of encoded operations, or does any finite amount of transmon dephasing preclude fault-tolerant operation of the encoded qubit?

I shall be happy to consider a revised manuscript that addresses the above questions.

Reviewer #3 (Remarks to the Author):

In their submission, "Implementing a Universal Gate Set on a Logical Qubit Encoded in an Oscillator," Heeres et al. demonstrate a high-fidelity gate set for a single qubit encoded using the cat-code onto a resonator mode. They demonstrate both a high-fidelity encoding and decoding process between a single transmon and this cat code, as well as high-fidelity gate operations designed using GRAPE and time-dependent drives while encoded into the cat code, verified with randomized benchmarking, encoded Wigner tomography, and decoded transmon spectroscopy.

The data are convincing, and clearly discussed. The ability to numerically design pulse sequences with GRAPE is not new (as the authors point out from its previous history in NMR); however, the use of such pulse sequences to manipulate single qubit information using a resonator cat-code is. The present manuscript thus presents an intriguing alternative to the usual microwave gate

operations performed at the level of a single transmon. To be useful, this technique will need to be scaled up to multiple qubits; the authors are aware of this extension, and indeed note its possibility in the conclusion. In my opinion, the manuscript is suitable for publication in Nature Communications.

Reviewer #1 (Remarks to the Author):

This paper describes the experimental implementation of a universal set of single qubit gate operations on a logical qubit, encoded in a superposition of even cat states of a microwave cavity. Such a logical qubit encoding has been called 'cat code' and in previous publications the authors have established that it can be used to correct for photon loss, the dominant decoherence mechanism in cavities. Compared to other encodings of logical qubits, cat codes require much less hardware resources and therefore appear very realistic.

The paper demonstrates experimentally the next step towards using logical cat-code qubits by demonstrating initialization from a transmon qubit and a universal set of single qubit gates. To achieve these operations the authors use optimal control strategies developed originally for NMR.

These are beautiful results, technically sound and well presented. Operating logical qubits is certainly a major step, which makes them, in principle, worth being published in Nature Communications.

Reply: *We of course appreciate that our work is perceived this way and in principle suitable for Nature Communications.*

However, while the reported gate fidelities of approximately 99% are sufficient for physical qubits as part of some error correction schemes, they are unacceptably low for a useful logical qubit. Presenting the results in the light of quantum error correction and logical qubits suggests that these gates are compatible with the cat code error correction schemes and benefit from them in terms of fidelity, but the authors do not provide any information supporting this suggestion nor do they emit any reservations. The current presentation of the paper in the light of quantum error correction is therefore misleading.

Reply: *When realizing a logical qubit one does not expect the logical operations on this qubit to be better than the ones on the underlying physical qubits right away. In fact, without any additional error correction the increased number of physical systems, or in our case number of levels, increases the effective decoherence rates. In our opinion, however, the experimental demonstration of high-fidelity operations on a logical qubit is still an important milestone in the field, and allows to evaluate the remaining sources of errors and non-idealities in real-world implementations.*

For the cat-code, logical operation errors implemented using our scheme will have a different character than the single-photon-loss errors the cat-code was designed to correct for. Therefore it will require a more complex protocol to correct for operation errors, which can for example be realized by embedding in an erasure-correcting code combined with error detection after an operation.

We feel these points were indeed not made clear enough in our previous version of the manuscript and have added a thorough treatment of these issues in the new Discussion section.

To remove this issue the authors should either clearly state that their gates cannot be protected with the cat-code error correction scheme and that their fidelity therefore has to be taken as is. Or they should give avenues how fidelities can be improved by correcting for the dominant error mechanisms. This will allow the reader to judge whether their approach is actually a huge step towards scalable quantum computation using cat codes, or just a beautiful side road.

I see at least two issues with correcting errors in these gates:

- The authors mention in their discussion of Table I that they can explain most of the infidelity by transmon dephasing. It is not obvious how this error could be corrected in the cat-code scheme, which would suggest that using a cat-code qubit does not present any benefits over directly using a transmon. Even worse, as the gates take much longer they may be affected even more by dephasing. Can the pulses be modified for the gates to be insensitive to transmon dephasing by adding appropriate cost functions to grape?

- It is not obvious that the resonator stays in a superposition of even cat states during the gate so the previously

proposed cat-code correction scheme might not work during the gate and it is not obvious if single photon loss during the gate can be corrected afterwards or is transformed into uncorrectable errors by the action of the drives. The authors might want to show resonator state evolution during the gate in the supplementary material.

Reply: *The cat-code error correction scheme can not be applied during the gates, as the system goes out of the logical subspace during these operations. However, single-photon loss is not a significant source of errors as transmon dephasing is indeed the dominant error mechanism. The errors introduced during the operations would need to be corrected by a higher level scheme, which we consider in the added discussion section regarding the nature of errors and how they can be detected and/or corrected.*

Note also that our logical qubit is much longer lived than our transmon, which is beneficial in several applications. The hybrid cat-state/transmon system would be useful in contexts in which both high-fidelity operations and long waiting times are required, for example for quantum repeaters.

Preliminary investigations revealed that adding cost functions to grape can minimize but not eliminate the effect of transmon dephasing, although this was not thoroughly explored. We have added figure S9 to the supplementary, which shows the oscillator state during each of the operations when starting from the logical +Z state.

In addition I have a few more minor issues:

- How does the scheme scale to larger cats which are required for the cat code correction scheme to be most effective?

Reply: *We are not quite sure what “scale to larger cats” really means. Scaling this error correction scheme would consist of applying a next layer of quantum error correction, using a different approach, to produce a second-level logical qubit, that would ideally be tailored to correcting the remaining errors from the first layer. In this implementation of the experiment, any non-orthogonality of the states due to the finite size of the cats is negligible and easily avoided. We have therefore not investigated how the scheme scales to larger cat sizes, α . In terms of quantum control of the system, we can generally say that operations involving larger photon numbers are more costly to optimize due to the larger Hilbert space. They are also harder to realize physically because they are more sensitive to accurate knowledge of the Hamiltonian and system transfer function. Again, however, the presently used size of the cat-states is enough to track and correct errors, as demonstrated in a recent paper by our group.*

- Figure 3b-c: The x axis labels should be identified more clearly

Reply: *We are slightly confused by this comment. There is no Figure 3c. The x-axis labels in Figure 3b are the elements of the process tomography, given in the standard Pauli operator basis. We have added a short statement to this effect in the figure caption.*

- The definition of fidelity should be given in the text when it is first used instead of the caption of table I.

Reply: *moved to main text*

Reviewer #2 (Remarks to the Author):

The authors describe experiments to demonstrate universal control of a protected logical qubit encoded in the state of a bulk microwave cavity dispersively coupled to a transmon qubit. The encoding scheme is the cat-code previously described by this group. Tools from optimal control theory are used to generate high-fidelity encoding and gate pulses; success of this approach requires accurate knowledge of higher-order terms in the qubit-cavity Hamiltonian. The experiments are beautiful and showcase an unprecedented level of control over an extremely complex multipartite quantum system. However, the current presentation raises obvious questions about the viability of the described approach as a means to realizing universal fault-tolerant quantum computation. It would be extremely useful for the community to have the authors address head-on some of the apparent limitations of the proposed approach, so that readers can judge how the cat-code approach to scalable quantum information processing with superconductors compares to more “conventional” approaches based on planar integrated circuits and the surface code.

Reply: *We thank the referee for these comments and for appreciating the fact that this experiment represents a significant advance in controlling the complex multi-level system which is the basis of the cat code. As we have pointed out in the new discussion section, this is a first paper on a road to producing logical qubits with higher fidelities through an alternative, hardware-efficient approach to error correction. In any implementation, be it the cat code or the better known stabilizer QEC or surface code, it may be some time before logical operations can be performed simultaneously with full error correction, all working with fidelities much better than the threshold in order to actually reduce the errors compared to a single physical qubit. Nonetheless, it is important and worthwhile to demonstrate the pieces and capabilities necessary to progress down this road.*

Indeed, in our experiment we can observe, quantify, and understand several of the current nonidealities of the implementation of the cat code. The surface code is one approach to error correction, but has not yet been experimentally realized or tested. When real experiments on a surface code can be performed, they will also need to demonstrate quite complex and accurate control over a complex system with many (perhaps even hundreds) qubits and a high-dimensional Hilbert space. In such an experimental realization, there will at first be many potential nonidealities or systematic errors, which could lead to errors which are not correctable in the surface code, either. Thus, we would expect that the first goal for any approach, including the surface code, is to place an actual real-world bound on these types of errors. We feel that the ability to address these kinds of questions for the cat code, today, and to begin the discussion of these issues in the community (with the added discussion section), is one of the main contributions of our results and this manuscript.

A full comparison of the potential strengths and weaknesses of the cat code versus the surface code or other approaches is, we feel, beyond the scope of this publication. The surface code and the cat code are different, but not necessarily competing, approaches. The surface code assumes that physical qubits are connected in a planar network but makes no assumptions about what the physical components are. The cat-code is designed to correct the errors in a specific physically motivated error model. Cat-encoded qubits could be integrated into a surface code as a higher level of encoding. In comparison with bare transmons on planar integrated circuits, cat-encoded qubits have lower operation fidelities but much higher lifetimes. Depending on the application, a varying amount of “wait time” may be needed, meaning one is not strictly preferable to the other. While this work does not present a fault-tolerant method of performing operations on cat-encoded qubits, it does not preclude it either. All current physical qubit implementations are susceptible to unwanted Hamiltonian terms (i.e. cross-talk, spurious couplings) which can induce uncorrectable errors.

Specifically:

1. Given that the fidelity of encoding in the cat code is less than the fidelity of gate operations on the bare (unprotected) transmon qubit, it would seem that operations in the protected subspace will always lead to lower fidelity than operations on the bare transmon, provided that one takes into account the penalty associated with encoding and decoding. Is there an understanding of the sources of error that contribute to infidelity of the encoding operations and a path to reducing this infidelity?

Reply: *In the new discussion section, we have addressed the issue of overhead and why fidelity may not improve immediately in the first successful implementation of logical operations. For any logical encoding, the increased system size will introduce effective decoherence larger than that of its physical components if no error-correction mechanism is applied. This also means that the fidelity of encoding, typically a complex operation itself, is generally worse than a single physical qubit operation. However, methods which can determine whether the encode operation successfully brought the system to the code-space could improve the the fidelity of logical state preparation. For example, one could measure the transmon after the encode operation to herald approximately 50% of the errors; the same holds for the logical operations. The fact that encode/decode fidelities predicted by numerical simulations agree with the experimentally determined value (table I) means that the dominant source of errors for these operations is well understood and the same as for the other operations: transmon dephasing.*

2. The cat-code is motivated by the observation that cavity photon loss is the dominant source of decoherence in this system; therefore, a code that allows one to monitor and correct this loss provides a high degree of protection. However, in their analysis of the fidelity of operations in the encoded subspace, the authors find that dephasing of the transmon is the dominant source of fidelity loss. A transmon dephasing time of 43 microseconds is quite respectable; at what point will transmon dephasing have negligible impact on the fidelity of encoded operations, or does any finite amount of transmon dephasing preclude fault-tolerant operation of the encoded qubit?

Reply: *As long as the transmon remains shorter lived than the linear oscillator, which seems likely to be the case, the lifetime of operations which completely entangle to two will be dominated by the transmon. It would be desirable that transmon decoherence leads to errors which are correctable. This might not be feasible in a system consisting of a single transmon and an oscillator, but it may be possible using a more complex ancilla system (such as another oscillator). However, such approaches are still under investigation, and at the moment we are not aware of a method to make the operations fault-tolerant.*

I shall be happy to consider a revised manuscript that addresses the above questions.

Reviewer #3 (Remarks to the Author):

In their submission, "Implementing a Universal Gate Set on a Logical Qubit Encoded in an Oscillator," Heeres et al. demonstrate a high-fidelity gate set for a single qubit encoded using the cat-code onto a resonator mode. They demonstrate both a high-fidelity encoding and decoding process between a single transmon and this cat code, as well as high-fidelity gate operations designed using GRAPE and time-dependent drives while encoded into the cat code, verified with randomized benchmarking, encoded Wigner tomography, and decoded transmon spectroscopy.

The data are convincing, and clearly discussed. The ability to numerically design pulse sequences with GRAPE is not new (as the authors point out from its previous history in NMR); however, the use of such pulse sequences to manipulate single qubit information using a resonator cat-code is. The present manuscript thus presents an intriguing alternative to the usual microwave gate operations performed at the level of a single transmon. To be useful, this technique will need to be scaled up to multiple qubits; the authors are aware of this extension, and indeed note its possibility in the conclusion. In my opinion, the manuscript is suitable for publication in Nature Communications.

Reviewer #1 (Remarks to the Author):

The authors have fully responded to my initial concerns. In particular, the added "discussion" section now helps non-expert readers appreciate these beautiful results in the context of quantum error correction with cat codes. I therefore recommend the revised manuscript for publication in nature communications.

Reviewer #2 (Remarks to the Author):

I have reviewed the revised manuscript of Heeres et al. I am satisfied with the authors' replies to the reviewer queries, and I very much like the added discussion in the main text regarding the path to fault tolerance, which properly situates this work in the context of other approaches to the realization of robust logical qubits. I recommend rapid publication of this manuscript in Nature Communications with no further revisions.